# Advances, challenges, and opportunities of human midbrain organoids for modelling of the dopaminergic system

Alessandro Fiorenzano [1,2,3✉], Edoardo Sozzi [2], Rahel Kastli[4,5], Maria Roberta Iazzetta[3,6], Andreas Bruzelius[2], Paola Arlotta[4,5] & Malin Parmar [2]

## Abstract

**Dopaminergic neurons in the ventral midbrain are critical for regulating movement, cognition, and emotion. Ventral midbrain organoids can be used to model both development and diseases of the dopaminergic system, especially Parkinson's disease. Here, we summarize recent advances and remaining challenges in developing such three-dimensional organoids from human pluripotent stem cells. We outline how ventral midbrain organoid systems have progressed from early three-dimensional culture models to sophisticated, engineered, multiregional systems that more accurately replicate the complex network of dopaminergic neurons. Furthermore, we examine how the development of organoid models from other brain regions, particularly the forebrain, provides complementary insights that can accelerate progress also in the field of midbrain organoids, towards the generation of more advanced in vitro systems for midbrain dopaminergic neurons and their circuitry. Such cutting-edge human stem cell-based models offer powerful platforms for investigating dopaminergic neuron generation, function, and connectivity, thereby enhancing disease modelling, drug discovery, and the development of targeted cell-based therapies.**

**Keywords** Human Pluripotent Stem Cells; Dopaminergic Neurons; Midbrain Development; Disease Modeling; Single-cell Sequencing
**Subject Categories** Development; Neuroscience; Stem Cells & Regenerative Medicine

## Introduction

The midbrain is located between the hindbrain (pons and medulla) and the forebrain (diencephalon) serving as a critical hub for sensory and motor signals. It is divided into ventral and dorsal regions. The ventral part of the midbrain is essential for motor control and cognition, and includes structures such as the substantia nigra (SN) and the ventral tegmental area (VTA). The dorsal part of the midbrain is primarily involved in sensory processing, particularly related to vision and auditory processing, and includes superior and inferior colliculi (Bjorklund and Dunnett, 2007; Fedtsova and Turner, 2001). This review focuses on three-dimensional (3D) models of dopaminergic (DA) neurons in the ventral midbrain (VM), the development and specification of which are largely guided by signals originating from the isthmic organizer and the floor plate (Fig. 1A). The isthmic organizer establishes the midbrain/hindbrain boundary by releasing morphogens along the anterior/posterior axis, while the floor plate controls dorso-ventral patterning of the neural tube, thereby defining the midbrain progenitor domain (Fig. 1A). Regulation of DA neural progenitor differentiation is fine-tuned by the combined action of several transcription factors, such as FOXA2 and LMX1A, which are critical for the development of mature DA neurons (Arenas et al, 2015). The ability to generate DA neurons from human pluripotent stem cells (hPSCs) offers a promising approach to recreating the architectural and molecular characteristics of VM tissue. Induced pluripotent stem cells (iPSCs) offer the unique advantage of studying patient-specific molecular dysfunctions associated with neurodegenerative disorders (Soldner et al, 2009; Wernig et al, 2008). This has allowed researchers to explore the complex etiology of these diseases, influenced by genetic variations, epigenetic backgrounds, and environmental factors (Bose et al, 2022). However, human stem cell-based models in two dimensions (2D) can be hampered by limited cellular interactions, failing to recapitulate the complex 3D environment of tissues in vivo. This can limit the neurons` ability to form proper synaptic networks in the absence of the tissue-specific microenvironment, and can hamper their functional maturation in long-term cultures (Liu et al, 2018; Quadrato and Arlotta, 2017).

Recent advances in 3D cellular models enable a more accurate recapitulation of human neurogenesis, leveraging the self-organizing and -patterning abilities of hPSCs (Lancaster et al, 2013; Quadrato et al, 2017). By integrating external patterning cues, it is now possible to generate brain organoids mimicking features of specific brain regions, such as VM organoids that demonstrate molecular identities and functional traits of the DA system (Fiorenzano et al, 2021a; Jo et al, 2016; Qian et al, 2016). Such models provide powerful platforms for investigating both the

[1]Department of Molecular Medicine and Medical Biotechnology, University of Naples "Federico II", Naples, Italy. [2]Department of Experimental Medical Science, Developmental and Regenerative Neurobiology, Wallenberg Neuroscience Center, Lund Stem Cell Center, Lund University, Lund, Sweden. [3]Stem Cell Fate Laboratory, Institute of Genetics and Biophysics "A. Buzzati-Traverso", CNR, Naples, Italy. [4]Department of Stem Cell & Regenerative Biology, Harvard University, Cambridge, MA, USA. [5]Stanley Center for Psychiatric Research, Broad Institute of MIT and Harvard, Cambridge, MA, USA. [6]Department of Precision Medicine, University of Campania Luigi Vanvitelli, Naples, Italy.
✉E-mail: alessandro.fiorenzano@unina.it

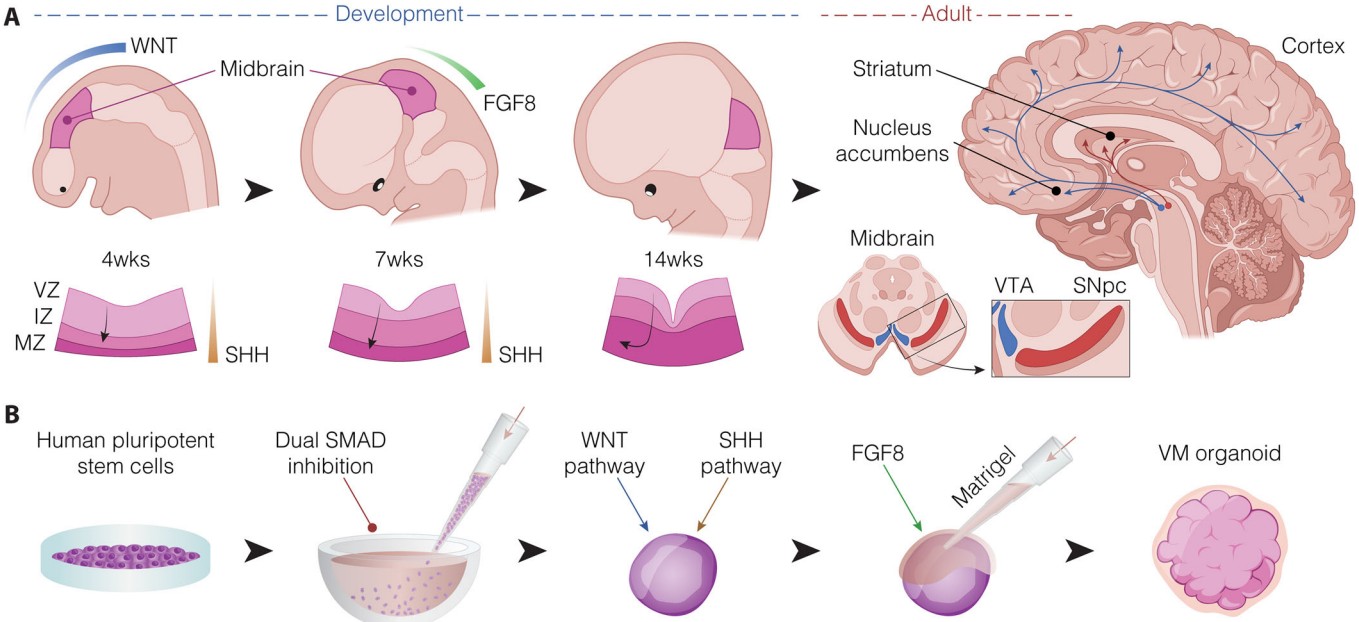

**Fig. 1. Generation of midbrain dopaminergic neurons in vivo and in vitro.**

(A) Schematic overview of human ventral midbrain (VM) during development and in the adult brain. The coronal section of the developing VM (bottom) is divided in three distinct areas: ventricular zone (VZ), intermediate zone (IZ), and mantle zone (MZ). Dopaminergic (DA) progenitors migrate first radially to the mantle zone and then tangentially while maturing into DA neurons, as depicted by black arrows. Gradients of key VM-determinants are represented by different colors (WNT blue, FGF8 green, SHH yellow). In the adult midbrain, DA neuron-populated nuclei are highlighted, as well as their projection patterns; medially the ventral tegmental area (VTA), housing A10 DA neurons projecting to the nucleus accumbens and cortical regions (in blue), and more laterally the substantia nigra pars compacta (SNpc), from where A9 DA neurons project to the dorso-lateral part of the striatum (in red). (B) Stepwise illustration of VM-patterned organoid generation. Homogeneous cultures of human pluripotent stem cells are first induced to neuronal fate through dual SMAD inhibition in low attachment conic wells. Patterning toward the VM is subsequently achieved through activation of WNT and SHH pathways and refined with exposure to FGF8. Embedding in extracellular matrix-mimicking substances, such as Matrigel, favors the maturation of VM organoids.

development and pathologies of human DA neurons, enabling detailed exploration of their normal functions as well as disease-related dysfunctions (Box 1). Furthermore, the advent of single-cell sequencing has offered unprecedented insights into human DA neurogenesis and facilitated the identification of novel molecular features and cell types (Box 2) (Fiorenzano et al, 2024; La Manno et al, 2016; Tiklova et al, 2019), which in turn has driven significant advancements in brain organoid technology (Pasca et al, 2019; Quadrato et al, 2017).

This review describes the state of the art, recent advances, and remaining challenges in modeling human DA neuron development and maturation using brain organoid technologies. It covers the evolution of 3D culture systems, from basic models to more advanced platforms capable of generating mature, functional DA neurons integrated within established neuronal circuits. It also focuses on how innovations in forebrain organoid models provide valuable insights that could accelerate the development of midbrain organoids for future applications (Box 3), particularly in the study of associated neurological diseases and the exploration of therapeutic strategies.

## Bridging in vitro and in vivo: the rise of human brain organoids

Over the past two decades, adherent two-dimensional (2D) culture systems have provided a foundational platform for stem cell differentiation protocols, offering controlled and reproducible

strategies for generating ventral midbrain dopaminergic neuronal lineages (Arenas et al, 2015; Chambers et al, 2009; Kriks et al, 2011; Ono et al, 2007). While inherently reductionist, 2D cultures offer several advantages: they support the production of homogeneous and highly pure populations of differentiated cells, ensure uniform exposure to patterning cues, and facilitate optimized gas exchange, all of which contribute to consistent and reliable differentiation outcomes (Nilsson et al, 2021; Nolbrant et al, 2017). Their amenability to genetic manipulation makes them particularly valuable for disease modeling and cell replacement strategies (Kirkeby et al, 2023).

In contrast, brain organoids harness the self-organizing potential of human pluripotent stem cells (hPSCs) to recapitulate key aspects of in vivo brain development. These three-dimensional (3D) systems follow intrinsic developmental trajectories that give rise to diverse neural and glial cell types within a spatially organized architecture (Fig. 1B) (Kadoshima et al, 2013; Lancaster and Knoblich, 2014). Importantly, brain organoids can be cultured over extended time periods, enabling the maturation of more complex and physiologically relevant neuronal circuits than is achievable in 2D models (Lancaster and Knoblich, 2014). As such, organoids serve as an intermediate model system, bridging the gap between simplified monolayer cultures and the full complexity of the human brain (Lancaster et al, 2013; Quadrato et al, 2017).

Initially, human brain organoids—largely relying on intrinsic patterning mechanisms—were primarily used to investigate early

---

**Box 1    Advancements in midbrain organoid technology: lessons from human forebrain models**

Recent advancements in midbrain organoid technology have shown promise for understanding DA neurogenesis and modeling neuronal function and disease. However, protocols for generating human midbrain organoids are still relatively new, with the first published after 2014 (Jo et al, 2016; Tieng et al, 2014). While this research area is expanding, it still presents significant limitations, as outlined in Table 1. Single-cell technologies have significantly improved culture protocols and the characterization of regionalized organoids (Box 2). Despite progress, further developments are needed, including refining culture conditions, a more comprehensive recapitulation of midbrain structures and cell types, and enhancing DA neuron maturation (Toh et al, 2023). Insights from the more established forebrain organoid field, with its advanced protocols and applications, could help accelerate improvements in midbrain organoid models for studying DA neurogenesis and related disorders (Birtele et al, 2023). Dorsal forebrain guided protocols were among the first guided protocols, using TGFβ, WNT, and BMP4 inhibition to induce forebrain cell development (Kadoshima et al, 2013; Mariani et al, 2015). These protocols generate neurogenic zones similar to the dorsal forebrain ventricular zone, producing cortical neurons in a spatial and temporal sequence resembling the endogenous tissue (Bershteyn et al, 2017; Qian et al, 2016). While these protocols reproduce key forebrain features, they often lack certain cells and structural complexity. Since then, protocols for other brain areas, including midbrain, cerebellum, and spinal cord, have emerged (Andersen et al, 2020; Atamian et al, 2024; Fiorenzano et al, 2021a; Muguruma et al, 2015). A key strategy to address missing interactions and cell types is the generation of "assembloids" by fusing organoids from different brain regions (Fig. 2B) (Pasca et al, 2022). For example, dorsal and ventral forebrain organoids were fused to create assembloids, enabling the study of inhibitory neuron migration (Bagley et al, 2017) (Birey et al, 2017). This

model has been used to investigate Timothy Syndrome (Birey et al, 2017) and has been expanded to study interactions across brain regions and with other systems (Andersen et al, 2020; Kasai et al, 2020; Miura et al, 2020; Xiang et al, 2019). Another major missing component in brain organoids is vascularization. While in 2D culture conditions nutrients are supplied through the media, in 3D organoid environments, diffusion limitations restrict nutrient and oxygen access to the deeper regions within the organoid core. While small, young organoids can sustain nutrient flow, larger organoids typically develop a hypoxic core made up of unspecified neurons, while the specified cell types are located along the periphery (Bhaduri et al, 2020; Uzquiano et al, 2022). Improving nutrient flow in organoids is crucial for modeling advanced brain development. One strategy is transplanting organoids into the rodent cortex, which promotes vascularization and integration with host sensory inputs, enhancing maturation (Mansour et al, 2018; Revah et al, 2022). This approach may also improve other organoid types. Forebrain organoids are valuable for studying early human cortical development and exploring complex, human-specific disorders using patient-derived stem cells. Studies have linked autism spectrum disorder risk genes with disruptions in early neurogenesis, underscoring the potential of forebrain organoids for investigating genetic disorders (Jourdon et al, 2023; Paulsen et al, 2022). A major limitation of organoid models is throughput and cost, especially for studying individual genetic backgrounds. Recent advancements have led to the generation of "chimeroids," multi-donor organoids that balance cell populations across donors (Fig. 2B) (Anton-Bolanos et al, 2024). This method has successfully demonstrated varied donor responses to stressors like ethanol and valproic acid, and shows promise for studying patient-specific susceptibilities in the midbrain.

---

**Box 2    Brain organoids and single-cell technologies—a powerful synergy**

The combination of brain organoid technology and single-cell sequencing is advancing rapidly, creating a powerful synergy that has the potential to transform experimental approaches in neurobiology. This integration allows researchers to explore cellular diversity and uncover underlying molecular mechanisms with unprecedented resolution (Fig. 3A) (Fleck et al, 2023). Advancements in next-generation sequencing allow detailed investigations of VM complexity at single-cell resolution across mammalian species and in 3D cultures (Fig. 3B) (La Manno et al, 2016; Tiklova et al, 2019). Single-cell sequencing has emerged as a crucial tool for deconstructing heterogeneous cell populations and identifying rare cell types in brain tissue, serving as reference atlases for in vitro systems (Braun et al, 2023; Han et al, 2018). This technology excels at distinguishing morphologically indistinguishable cells by elucidating distinct molecular identities, which reveal varying degrees of developmental potential and specialized functions in both developing and adult brains (Fig. 3B) (Eze et al, 2021). Furthermore, cutting-edge single-cell sequencing integrates multimodal measurements, combining genetic, epigenetic, and transcriptomic data from individual cells (Fig. 3A) (Welch et al, 2019; Zhu et al, 2020). Multi-omics approaches facilitate simultaneous profiling of the epigenome, transcriptome, and proteome across thousands of individual cells, thus enabling the exploration of fundamental processes that govern brain function and disease (Angermueller et al, 2016; Luo et al, 2022). The interplay between epigenetic factors and intrinsic/extrinsic influences shapes the developmental potential and plasticity of stem cells, making the integration of multi-omics and single-cell data particularly valuable for addressing key questions in VM development (Fig. 3A,B). The integration of single-cell sequencing with brain organoid technology marks a crucial step toward fully elucidating the roles of key regulators that orchestrate the patterning of cells into distinct brain tissue identities, which may otherwise remain unexplored.

neurodevelopment and tissue organization (Kanton et al, 2019). These early protocols, often referred to as "unguided," are characterized by limited use of exogenous signals, relying instead on the innate self-patterning capabilities of hPSCs (Pasca et al, 2022). However, recent advances have introduced "guided" differentiation strategies that apply extrinsic patterning factors to promote the specification of distinct brain regions such as the cortex, striatum, cerebellum, and midbrain (Atamian et al, 2024; Fiorenzano et al, 2021a; Jo et al, 2016; Kanton et al, 2019; Miura et al, 2020; Qian et al, 2016; Velasco et al, 2019). These protocols offer enhanced precision and reproducibility, enabling the generation of regionally defined organoids with consistent cytoarchitecture and cellular composition (Box 1).

Notably, guided midbrain organoid models can now recapitulate the spatial layering and molecular features of the developing human VM, faithfully mimicking aspects of dopaminergic neuron development observed in vivo (Jo et al, 2016; Qian et al, 2016; Renner et al, 2020). As such, region-specific organoid systems offer a powerful platform for modeling discrete brain areas and provide a physiologically relevant context for studying neurodevelopmental processes, neurodegeneration, and psychiatric disorders (Renner et al, 2020).

# Using ventral midbrain organoids to model midbrain development

State-of-the-art methods for generating midbrain organoids from hPSCs involve the inhibition of TGF-β and BMP signaling at early differentiation stages through modulation of their effectors, such as

**Box 3  Questions and future directions**

Although substantial progress has been made in modeling DA neurons´ functions and dysfunctions using brain organoids, several scientific and technical challenges remain. To fully harness the potential of these models for advancing our understanding of human brain development and disease, it is critical to identify current limitations and strategically direct future research. Key questions and priority areas include:

- Model fidelity and maturation: To what extent do current brain organoids recapitulate in vivo developmental trajectories and cellular diversity? What strategies can enhance DA neuron maturation and subtype identity?
- Circuitry and functional integration: How can synaptic connectivity and network-level activity be more accurately reconstructed and functionally assessed in organoid systems?
- Standardization and reproducibility: How can protocols be harmonized across laboratories and cell lines to reduce variability and improve the reproducibility of organoid-based findings?
- Disease modeling and translational relevance: What are the limitations of current iPSC-based disease models, and how can tools such as CRISPR/Cas9 enhance their mechanistic insight and clinical applicability?
- Scalability and technological integration: What innovations in bioengineering, imaging, and computational modeling are needed to increase organoid complexity, scalability, and physiological relevance—particularly through integration with vasculature, immune components, or other brain regions?

**Future research should prioritize:**

- Developing organoid systems integrated in functional neuronal circuitries.
- Applying multi-omic profiling and live imaging to better characterize developmental and disease-associated processes.
- Establishing novel platforms for high-throughput drug screening and personalized medicine using patient-derived organoids.
- Fostering collaborative efforts to define benchmarks and develop open-access datasets for model comparison and validation.

the SMAD proteins, to drive neural differentiation (Qian et al, 2016). The ventralizing factor sonic hedgehog (SHH), combined with canonical WNT signaling, establishes VM regional identity by inducing LMX1A$^+$/FOXA2$^+$/OTX2$^+$ floor plate progenitor cells (Jo et al, 2016; Renner et al, 2020). These progenitors then differentiate in a 3D environment into functionally mature DA neurons during long-term culture (Fig. 1A). Rostro-caudal patterning of VM progenitors is achieved through timed delivery of FGF8b, which mimics its release from the midbrain-hindbrain boundary (Fig. 1A; Table 1) (Fiorenzano et al, 2021a). The resulting midbrain organoids successfully establish distinct layers of neuronal cells in vitro, expressing markers characteristic of developing human VM, similar to the in vivo development of DA neurons (Figs. 1B and 2A) (Fiorenzano et al, 2021a; Jo et al, 2016; Qian et al, 2016). After an extended period of time in culture, DA neurons within these organoids start to produce neuromelanin-like granules, comparable to those found in the human *substantia nigra pars compacta* (Fiorenzano et al, 2021a; Sozzi et al, 2022b). This indicates that functionally mature DA neurons develop within VM organoids and are organized into distinct populations, including neuromelanin-containing DA neurons. Furthermore, integrating single-cell technologies with human VM organoids has revealed molecularly distinct subtypes of human DA neurons, including A9

and A10 subtypes (Fiorenzano et al, 2024; Poulin et al, 2020; Poulin et al, 2014), which are linked to motor control and cognitive function, respectively (Bjorklund and Dunnett, 2007; Garritsen et al, 2023) (Box 2). However, differences between cells in vitro and in vivo present significant challenges in establishing definitive standards for the authenticity of developing and mature human DA neurons in culture. To address this limitation, a recent study generated 3D organoid-like structures from the fetal VM of human embryos aged 6.5–11 weeks post-conception (Birtele et al, 2022). By employing single-cell RNA sequencing, the authors extensively analyzed the molecular profiles of the cell types maturing within these fetal 3D cultures. This comprehensive dataset provides a valuable reference for defining authentic human DA neurons in culture (Box 3), offering crucial insights and benchmarks for comparing hPSC-derived DA neuron models. Despite significant progress, many aspects of DA neuron development and subtype-specific maturation remain unclear. A key research area is represented by the identification and characterization of DA neuron subtypes, their molecular signatures, and how these distinctions contribute to functional diversity (Fiorenzano et al, 2024; Poulin et al, 2020). Unraveling the mechanisms that drive DA subtype specification during development and from stem cells is crucial, as it could pave the way for more precise therapeutic strategies for Parkinson's disease and other disorders affecting the DAergic system.

The direct comparison between hPSC-derived and fetal DA tissue not only highlights similarities and differences in DA neurons, but also underscores the presence of other essential cell types in fetal tissue that are not present in VM organoids (Birtele et al, 2022; La Manno et al, 2016). Moreover, for DA neurons to function effectively, they must establish functional synaptic connections within appropriate neural circuits (Garritsen et al. 2023). Synaptogenesis, including the formation of DA-releasing synapses and their integration into broader neural networks, is fundamental for midbrain functionality (Bjorklund and Dunnett, 2007). Beyond DA neurons, the midbrain contains non-DA populations, such as GABAergic interneurons, glial cells (astrocytes, oligodendrocytes, and microglia), and glutamatergic neurons, all of which play critical roles in modulating DA neuron activity and maintaining circuit homeostasis (Ikai et al, 1992; Nair-Roberts et al, 2008). Glial cells are thought to be particularly important for supporting DA neuron differentiation and survival. They release key factors that promote the differentiation of neural progenitors into DA neurons, including glial cell line-derived neurotrophic factor (GDNF)—a crucial molecule for DA neuron maturation and maintenance within the midbrain (Lin et al, 1993). While astrocytes are well-represented in hPSC-derived VM organoids, microglia remain absent, limiting the ability of current models to fully replicate the VM's native cellular diversity (He et al, 2025). The cellular microenvironment is critical for DA neuron maturation, with neuron-glia interactions and blood–brain barrier components shaping their development and stability. Advanced organoid models integrate extracellular matrix components, vascularization, and glial support to offer a more physiologically relevant platform (Bagley et al, 2017; Bhaduri et al, 2020; Birey et al, 2017; Uzquiano et al, 2022). Proper differentiation and integration of these diverse cell types are also essential to constructing a more complete and functional midbrain model (Mansour et al, 2018; Revah et al, 2022) (Box 3).

**Table 1. Summary of ventral midbrain organoid protocols developed to date.**

| Preliminary steps/cell preparation | Seeding cell for 3D structure | Early patterning factors | Polymer embedding and spinning culture | Late patterning and maintenance factors | Reference |
|---|---|---|---|---|---|
| / | hPSCs (1000 cells per micro-well) | LDN 0.5 µM<br>SB 10 µM<br>SHH 100 ng/mL<br>FGF-8 100 ng/mL<br>PMA 2 µM<br>CHIR 3 µM | No polymer embedding<br>Orbital shaker from day 1 | cAMP 0.5 mM<br>AA 0.2 mM<br>GDNF 10 ng/mL<br>BDNF 10 ng/mL<br>TGFβ3 1 ng/mL<br>FGF-20 5 ng/mL<br>Trichostatin A 20 nM<br>Compound E 1 µM<br>DAPT 10 µM | Tieng et al, 2014 |
| / | hPSCs (10,000 cells per well) | SB 10 µM<br>NOGGIN 200 ng/mL<br>CHIR 0.8 µM<br>SHH 100 ng/mL<br>FGF8 100 ng/mL | Day 7, Matrigel<br>Orbital shaker from day 7 | BDNF 10 ng/mL<br>GDNF 10 ng/mL<br>AA 100 µM<br>db-cAMP 125 µM | Jo et al, 2016 |
| / | hiPSCs | SHH 100 ng/mL<br>PMA 2 µM<br>FGF-8 100 ng/mL<br>SB 10 µM<br>LDN 100 nM<br>CHIR 3 µM | No polymer embedding<br>SpinΩ Bioreactor from day 14 | TGF-β 1 ng/mL<br>cAMP 0.5 mM<br>BDNF 20 ng/mL<br>GDNF 20 ng/mL<br>AA 0.2 mM | Qian et al, 2016 |
| **hNESCs from hiPSCs** (Reinhardt et al, 2013) | hNESCs (9000 cells per well) | CHIR 3 µM<br>PMA 0.75 µM<br>AA 150 µM | Day 8, Matrigel<br>Orbital shaker from day 14 | BDNF 10 ng/mL<br>GDNF 10 ng/mL<br>db cAMP 500 µM<br>AA 200 µM<br>TGF-β3 1 ng/mL<br>PMA 1 µM | Monzel et al, 2017 |
| / | hiPSCs | CHIR 3 µM<br>A83-01 0.5 mM<br>LIF 10 ng/mL<br>FGF-2 20 ng/mL<br>EGF 20 ng/mL | Day 0, Matrigel<br>Orbital shaker from day 0 | AA 200 nM<br>BDNF 20 ng/mL<br>GDNF 20 ng/mL<br>SHH 100 ng/mL | Kim et al, 2019 |
| **mfNPCs from hiPSCs** SB 10 µM,<br>LDN 150 nM<br>CHIR 3 µM,<br>SAG 0.5 µM<br>AA 200 µM | mfNPCs (3000 cells per well) | SAG 0.5 µM<br>CHIR 0.7 µM<br>AA 200 µM | / | BDNF 10 ng/mL<br>GDNF 10 ng/mL<br>AA 200 µM<br>db cAMP 500 µM<br>TGF-β3 1 ng/mL<br>Activin A 2.5 ng/mL<br>DAPT 10 µM | Smits et al, 2019 |
| / | hPSCs (40 × 10^6 cells per flask) | SB 10 µM<br>LDN 100 nM<br>PMA 2 µM<br>SAG 1 µM<br>CHIR 1 µM | / | BDNF 10 ng/mL<br>GDNF 10 ng/mL<br>AA 0.2 mM<br>db cAMP 0.1 mM<br>DAPT 10 µM | Ahfeldt et al, 2020 |
| / | hPSCs (10,000 cells per well) | Dorsomorphin 2 µM<br>A83-01 2 µM<br>SAG 2 µM<br>FGF8 100 ng/mL<br>CHIR 3 µM | Day 4, Matrigel<br>Orbital shaker from day 9 | BDNF 10 ng/mL<br>GDNF 10 ng/mL<br>AA 200 µM<br>db cAMP 125 µM | Kwak et al, 2020 |
| **smNPCs from hiPSCs** (Reinhardt et al, 2013) | smNPCs (9000 cells per well) | SAG 0.5 µM<br>CHIR 3 µM<br>AA 200 µM<br>BDNF 1 ng/mL<br>GDNF 1 ng/mL | / | BDNF 2 ng/mL<br>GDNF 2 ng/mL<br>AA 200 µM<br>TGFβ − 3 1 ng/mL<br>db cAMP 100 µM<br>Activin A 5 ng/mL (day 6 only) | Renner et al, 2020 |
| / | hPSCs (40 × 10^6 cells per flask) | SB 10 µM<br>LDN 100 nM<br>PMA 2 µM<br>SAG 1 µM<br>CHIR 1.5 µM | No polymer embedding<br>Orbital shaker from day 22 | BDNF 20 ng/mL<br>GDNF 20 ng/mL<br>AA 0.2 mM<br>dcAMP 0.1 mM<br>DAPT 10 µM | Sarrafha et al, 2021 |

**Table 1.** (continued)

| Preliminary steps/cell preparation | Seeding cell for 3D structure | Early patterning factors | Polymer embedding and spinning culture | Late patterning and maintenance factors | Reference |
|---|---|---|---|---|---|
| hNESCs from hiPSCs (Reinhardt et al, 2013) | hNESCs (9000 hNESCs per well) | AA 150 µM<br>CHIR 3 µM<br>PMA 0.75 µM | Day 6, Matrigel or Geltrex Horizontal shaker from day 14 | BDNF 10 ng/mL<br>GDNF 10 ng/mL<br>AA 200 µM<br>TGFβ3 1 ng/mL<br>db cAMP 500 µM<br>PMA 1 µM (for 6 days) | Zagare et al, 2021 |
| / | hPSCs (8000 hPSCs per well) | SB 10 µM<br>Noggin 150 ng/mL<br>SHH 400 ng/mL<br>CHIR 1.5 µM<br>FGF8b 100 ng/mL | Day 14, Matrigel Orbital shaker | BDNF 20 ng/mL<br>GDNF 10 ng/mL<br>AA 200 µM<br>db-cAMP 500 µM<br>DAPT 1 µM | Sozzi et al, 2022b 2024 |
| / | hPSCs (300–750 cells per micro-well) | SB 10 uM<br>LDN 100 nM<br>CHIR 0.6 µM<br>SAG 400 nM<br>FGF8b 100 ng/mL<br>LM22A-4 2 µM<br>AA 200 µM | No polymer embedding Orbital shaker | LM22A-4 2 µM<br>AA 200 µM<br>GDNF 10 ng/mL<br>dcAMP 500 µM<br>DAPT 1 µM | Chen et al, 2023 |

smNPC, small molecule neural precursor cells; NESCS, Human neuroepithelial stem cells; mfNPCs, floor plate neural progenitor cells; LDN-193189, TGF-beta/Smad inhibitor; LIF, Leukemia inhibitory factor; SAG, Shh Signaling Agonist; PMA, purmorphamine, Shh signaling agonist; ActivinA, member of TGF-beta family; LM22A-4, BDNF agonist; A83-01, SMAD inhibitor; EGF, epidermal growth factor.

# Disease modeling using ventral midbrain organoids

Parkinson's disease (PD) is the second most common neurodegenerative disorder, affecting ~2–3% of individuals aged 65 and above. PD is characterized by the progressive loss of DA neurons in the SN, and the main symptoms include bradykinesia, tremor, rigidity, and postural instability (Bjorklund and Dunnett, 2007). While current therapies can effectively address motor symptoms for several years, there is no available treatment for regenerating the brain or halting the progression of the disease. There is growing interest in employing human cell culture systems to advance our comprehension of PD etiology and progression (Fiorenzano et al, 2021b). Such models include those derived from human embryonic stem cells (hESCs), patient-derived neurons, iPSCs, brain organoids, and postmortem tissue (Fig. 2A) (Vadodaria et al, 2020). Serving as tools for in vitro analysis of disease mechanisms, brain organoids can have an important function to bridge the existing gap between experimental research and clinical application. Building on the success of forebrain organoids, 3D VM cultures have seen a significant rise in their use for modeling both DA neuron development and PD in recent years. A key factor driving these advances in organoid models for human disease has been the revolutionary introduction of cell reprogramming technologies (Takahashi and Yamanaka, 2006), which enable the generation of iPSCs directly from PD patients (Soldner et al, 2009). These iPSCs can then be differentiated into DA neurons, offering a powerful platform for disease modeling (Hiller et al, 2022; Laperle et al, 2020). The significance of this approach is exemplified by studies demonstrating that midbrain-specific organoids derived from PD patients carrying the LRRK2-G2019S mutation are capable of recapitulating key disease-relevant phenotypes. In a recent study, isogenic 3D midbrain organoids with or without the PD-associated *LRRK2*-G2019S mutation were generated, successfully reproducing brain pathological hallmarks in vitro (Kim et al, 2019). These

mutant organoids displayed heightened susceptibility to induced neurotoxic damage, leading to increased apoptosis. Notably, phosphorylated α-Syn was localized in endosomes, and an increase in mitophagy was observed. The reduction in expression of a specific thiol-oxidoreductase, TXNIP, significantly mitigated aggregated α-Syn (Kim et al, 2019). Another study investigated the diminished number and complexity of midbrain DA neurons in *LRRK2*-G2019S mutant organoids compared to controls (Smits et al, 2020). Expression of the floor plate marker FOXA2, crucial for midbrain DA neuron generation, was elevated in PD patient-derived midbrain organoids, implying a neurodevelopmental defect in midbrain DA neurons expressing *LRRK2*-G2019S (Smits et al, 2020). Importantly, iPSCs-derived organoids that allow for long-term culture can be potentially used to investigate not only the role of known genetic mutations associated with PD, but also sporadic cases, which are accounting for more than 90% of the total PD diagnosis and are more difficult to recapitulate with current models (Dawson et al, 2010; Vadodaria et al, 2020). However, the distinct genetic backgrounds of iPSC-derived organoids, intrinsic organoid-to-organoid variability (Velasco et al, 2019), and the rejuvenation of cells during iPSC reprogramming remain significant challenges, playing a crucial role in shaping their development and response to external factors (Volpato and Webber, 2020).

Currently, VM organoids have primarily been used to model PD (Box 3). However, a few studies have expanded their use to model other neurodegenerative diseases. For instance, VM organoids have been employed to model Progressive Supranuclear Palsy (PSP), a severe neurodegenerative condition characterized by the accumulation of hyperphosphorylated tau protein and tufted astrocytes (Parrotta et al, 2025). In addition, Huntington's Disease (HD) has been modeled by fusing human VM organoids with striatum-patterned organoids established from HD patient-derived iPSCs (Wu et al, 2024). Regarding the use of VM organoids to model neuropsychiatric disorders in which the midbrain plays an important role, we found limited data (Abbott et al, 2023).

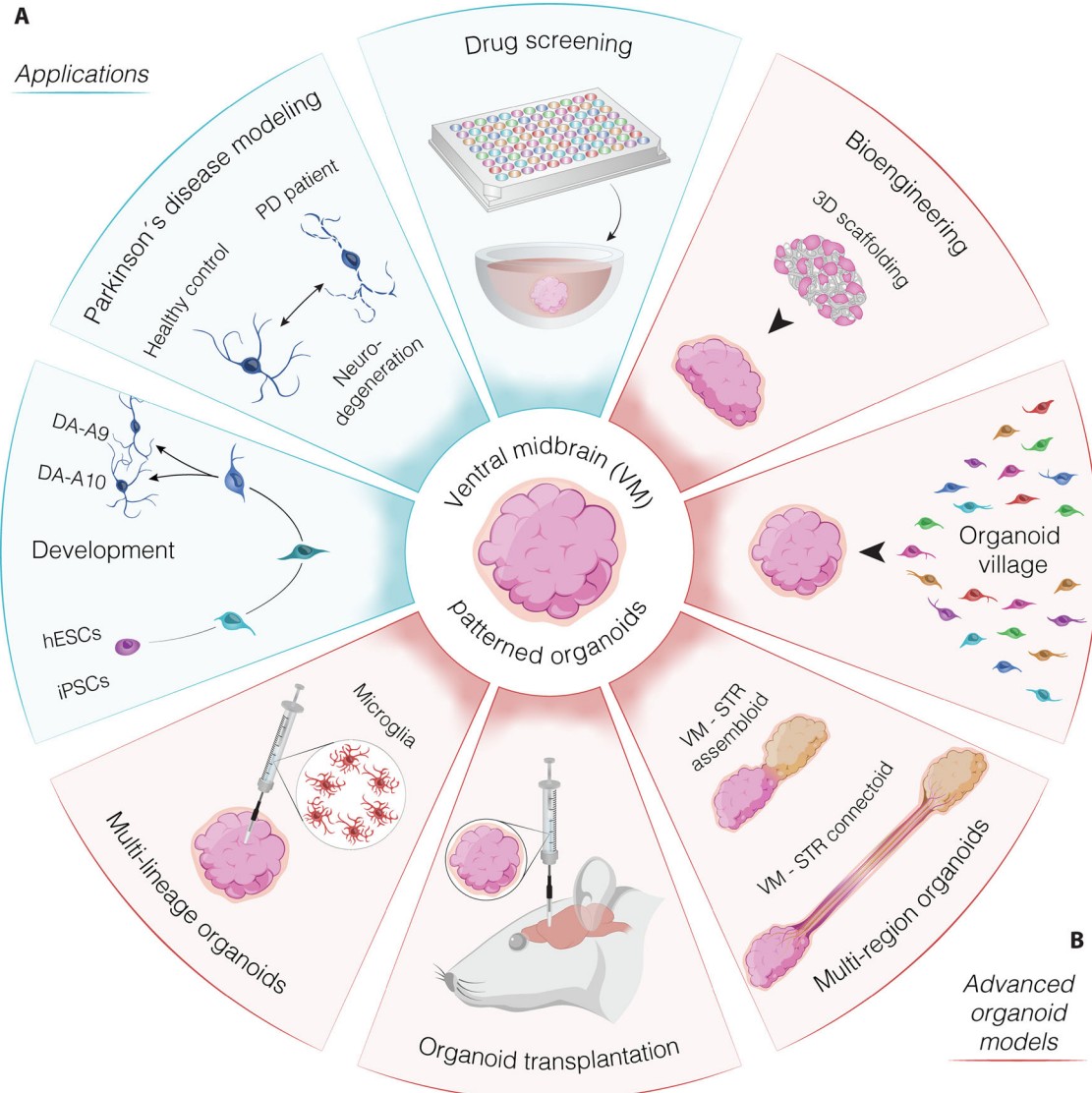

**Fig. 2. Advancements and applications of ventral midbrain organoid models.**

(A) VM organoids are primarily used to study the development of DA neurons derived from hPSCs or iPSCs, model Parkinson's Disease (PD)-related features, and for drug screening applications. (B) Advanced organoid-based models incorporate bioengineering techniques and multilineage or multi-region approaches, such as integrating VM organoids with striatal (STR) patterned cultures to better mimic brain circuitry.

This could be largely attributed to the difficulty in generating sub-regionalized VM organoids necessary to model such dysfunctions.

While VM organoids provide a powerful platform for studying midbrain development, their limitations must be considered when modeling age-dependent neurodegenerative disorders. PD, for instance, is a progressive condition influenced by aging, cellular stressors, and neuroinflammation—factors that are not fully represented in standard VM organoid models. Despite these constraints, VM organoids can still provide valuable insights into PD pathogenesis, particularly in understanding genetic predispositions and early cellular dysfunctions. These models may allow for high-content screening of small molecules or gene therapies aimed at rescuing these phenotypes before neurodegeneration occurs (Kim et al, 2019; Smits et al, 2020).

To expand their relevance for PD research, VM organoids require further refinements to better recapitulate age-related pathological features. This could involve other strategies for reprogramming, such as direct conversion where molecular features linked to the age of the donor are maintained (Jullien et al, 2017; Mertens et al, 2015) and that have been shown to recapitulate disease pathology of genetic and sporadic PD (Drouin-Ouellet et al, 2022; Victor et al, 2020). Other potential strategies include extended culturing, exposure to pro-aging factors, or co-culture with microglia, incorporating key aspects of neuroinflammation and oxidative stress that drive PD progression. Additionally, bioengineering approaches integrating vascularization and extracellular matrix components could enhance neuronal maturation and metabolic support, making organoids more representative of adult, and aged, midbrain physiology.

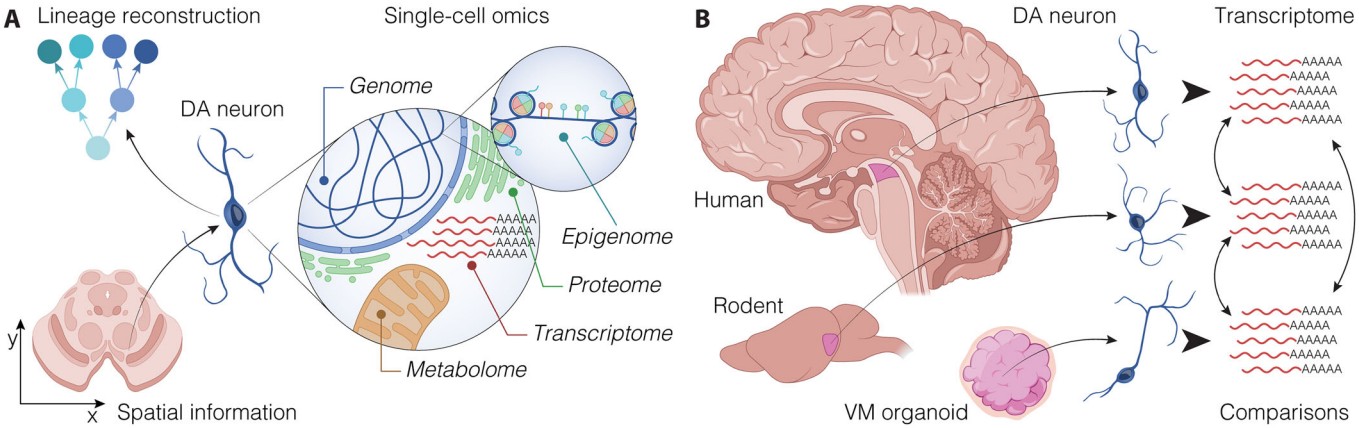

**Fig. 3. Resolving dopaminergic systems with single-cell technologies.**

(A) Single-cell omics (transcriptomics, metabolomics, proteomics) enable detailed profiling of DA progenitors and neurons, with advances like spatial integration and lineage reconstruction via inheritable barcodes. (B) DA neurons derived from VM-patterned organoids, human, or rodent brains allow for transcriptomic comparisons.

# Drug screening in ventral midbrain organoids

Neurodegenerative disorders, including PD, often exhibit complex and sporadic etiologies. Animal models carrying familial mutations may not always allow the study of key therapeutic targets, despite reflecting some disease phenotypes (Dovonou et al, 2023; Pingale and Gupta, 2020). Psychiatric and cognitive diseases, such as depression, autism spectrum disorder, and schizophrenia, pose challenges due to the complexity of patients' biological traits, making them difficult to model in rodents. In addition, non-human primate models, while exhibiting more human-like behavioral patterns, are cost-prohibitive for generating the numerous samples needed for high-throughput screening programs and involve significant ethical considerations (Park et al, 2024). In the field of drug screening, the use of brain organoids that reflect the growth environments of cells in more relevant physiological conditions is now emerging as a more viable approach compared to 2D culture, which presents limitations in recapitulating cell–cell interactions and complex cytoarchitectures (Fig. 2A). As described previously, iPSCs can be used to generate patient-specific organoids to model not only neurodevelopmental but also neurodegenerative diseases (Vadodaria et al, 2020). In the context of PD, the standard treatment involves the administration of DA replacement therapies, with L-Dopa being the most prevalent option. However, chronic use of these medications can lead to several complications, including dyskinesia, motor fluctuations, and cognitive impairments (Gustavsson et al, 2011; Obeso et al, 1989; Salat and Tolosa, 2013). These challenges highlight the urgent need for innovative therapies that offer disease modification, enhanced efficacy and/or reduced side effects. The advent of human-derived models, particularly VM organoids, presents a promising platform for drug discovery and screening (Fig. 2A). Such organoids provide a more physiologically relevant environment for evaluating drug responses and mechanisms of action. Moreover, leveraging organoids generated from patient-specific iPSCs could facilitate personalized medicine approaches, allowing for tailored treatment strategies that cater to the unique biological profiles of individual patients with PD

(Smits et al, 2019; Vadodaria et al, 2020). In addition to their applications in neurodegenerative diseases, organoid models show significant promise for drug screening in neuropsychiatric disorders, particularly given the critical role that dopaminergic neurons play in cognitive function. Drug efficacy assessment in these patients is often challenging, underscoring the need for effective screening methodologies. Conditions such as schizophrenia, bipolar disorder, and major depressive disorder present complex clinical challenges, largely due to the variability in patient responses to medications and the difficulties in accurately measuring treatment efficacy (Readhead et al, 2018). Renner and colleagues pioneered the development of midbrain organoids with key features resembling the human midbrain and its spontaneous neural activity adaptable for drug screening (Renner et al, 2020). The key innovation lies in the complete automation of the workflow, from generation to analysis, resulting in increased reproducibility within and across batches. This advancement was validated through comprehensive techniques such as RNA sequencing and whole-mount high-content quantitative imaging (Renner et al, 2020) (Box 2). The automation not only streamlines processes but also enables the precise assessment of drug effects at single-cell level within the intricate 3D cellular environment, all integrated into a fully automated high-throughput screening workflow (Fig. 3A; Box 3).

# Challenges and progress in midbrain organoid research

## Bioengineered approaches for ventral midbrain organoid cultures

Scaffold-supported brain organoids have emerged as a promising approach for generating highly mature and viable neurons (Lancaster et al, 2017; Sozzi et al, 2022a; Tejchman et al, 2020) (Fig. 2B). Medical-grade carbon fibers, introduced as a novel scaffold for organoid culture, demonstrated enhanced efficiency in hPSC DA neuron differentiation within organoids compared to the previously

tested poly-lactic-co-glycolic acid (PLGA) copolymer scaffold (Tejchman et al, 2020). The favorable physicochemical properties of carbon scaffolds, including porosity, microstructure, and stability in the cellular environment, make them a suitable structure for creating in vitro VM organoid models (Tejchman et al, 2020). The use of silk microfibers as scaffolds has also proven to enhance the functional properties of VM organoids (Fiorenzano et al, 2021a). Silk scaffolding resulted in the uniform generation of more mature DA neurons, spanning both the core and outer layers of bioengineered VM organoids. Spinning bioreactors have also been widely utilized to enhance nutrient and oxygen diffusion within brain organoids by agitating and improving medium circulation, as the limited contact surface of the organoid with the culture medium leads to inner core cell death at later stages of culture. Moreover, bioengineering and biofabrication platforms have also contributed to the advancement of brain organoid culture. Polydimethylsiloxane (PDMS) soft lithography, a standard method for microfluidic system fabrication, was employed to create platforms that enhance organoid viability (Cho et al, 2021; Khan et al, 2021; Qian et al, 2016; Rezaei et al, 2023; Seiler et al, 2022. Miniaturized spinning bioreactors, created through 3D printing, have successfully decreased culturing media volume and incubator space compared to commercial spinning bioreactors, facilitating the long-term maintenance of brain organoids (Khan et al, 2021, #7972). A recent study described a 3D printing pipeline established for fabricating tailor-made culturing platforms designed for spatially separated, but connected, brain organoid array cultures (Rezaei et al, 2023). This all-in-one platform expedites all culturing steps (including cellular aggregation, spheroid growth, hydrogel embedding, and VM organoid maturation) within a single well plate, without the need for organoid manipulation or transfer (Rezaei et al, 2023). Similarly, a cutting-edge multisensory integrated organ-on-a-chip platform, monitoring electrophysiology, gas exchange, and dopamine release in human midbrain organoids, showed that microfluidic culture substantially diminished necrotic core formation (Spitz et al, 2024). These advancements not only enhance the efficiency of DA tissue generation but also present a novel system for PD modeling.

## Generation of multilineage ventral midbrain organoids

Neuronal function is intricately shaped by interactions with various cell types, such as astrocytes, oligodendrocytes, endothelial cells from the surrounding vasculature, and microglia (La Manno et al, 2016) (Cserep et al, 2021; Tiklova et al, 2020). The VM is the result of a complex assembly of cell types derived from the neuroectoderm (neurons, astrocytes, oligodendrocytes), yolk sac (microglia), and mesoderm (vascular cells). As VM organoids are generated exclusively from progenitors of neuroectodermal origin, microglia and functional vascular cells are absent in such models, and this poses limitations in recapitulating the brain environment. For example, microglia, vital for immune regulation in the brain, play a significant role in neuroinflammation, immune response, and influence the vulnerability of neurons and progression of neuronal death in neurodegenerative disorders, as microglia contribute to network remodeling and phagocytosis (Lin et al, 2021; Tremblay et al, 2019; Ueta and Miyata, 2021; Zhang et al, 2023). Challenges persist in integrating diverse germ layer-derived components such as blood vessels and microglia into existing brain

organoid systems, hindering a comprehensive understanding of the complex cellular interactions within the brain. However, recent advancements include the fusion of cerebral vessel organoids with brain organoids, resulting in the formation of blood–brain barrier-like structures and an abundance of microglia, thus offering a more complete model for studying neurodegenerative processes (Adams et al, 2023; Bergmann et al, 2018; Sun et al, 2022) (Fig. 2B; Box 1). Ongoing efforts are therefore focused on refining induction conditions to achieve stable and mature organoid development, enhancing their suitability for investigating brain development, neurodegenerative diseases such as PD, and neuroinflammation. Soler and colleagues recently conducted a study where hPSCs were differentiated into microglia, and subsequently injected into VM organoids (Sabate-Soler et al, 2022). Using single-nucleus RNA sequencing, they characterized the injected microglia and explored their impact on other cell types within midbrain organoids (Box 2). The study demonstrated that microglia play a central role in synaptic remodeling, contributing to increased activity of DA neurons. These advanced studies aim to establish a more suitable system for in-depth exploration of brain development and neurodegenerative diseases, including the interaction of other cell types with VM cells in a fully humanized, easily accessible and scalable system (Sabate-Soler et al, 2022). At the same time, other multilineage organoid models are being developed to address important biological questions involving the interaction of the VM with other organs in the human body (Reiner et al, 2021). For example, it was recently observed that accumulated α-Syn protein can be found in the gastrointestinal tract in early stages of PD, highlighting a bidirectional communication in the brain–gut axis (Elfil et al, 2020). The brain exerts influence on intestinal activity and function, while the gut microbiome plays a role in maintaining gut epithelium mucus, and its metabolites impact the immune system and brain function. Notably, PD patients manifest a distinct microbiota composition, along with altered concentrations of short-chain fatty acids and various cytokines, indicating immune system involvement (Matheoud et al, 2019; Sampson, 2020). The instructive role of intestinal microbiota in PD has been known for a long time, contributing to motor deficits, microglia activation, and α-Syn pathology. Exploiting these insights, brain organoids derived from PD patients have significantly advanced our comprehension of the disease pathophysiology. However, to fully grasp the multifaceted nature of PD, there is a compelling need to enhance the complexity of these models. Future studies might explore the role of microbiota-derived metabolites on organoid systems, thus mimicking the gut–brain axis in PD.

## Brain organoid transplantation

The absence of blood vessels and microglial cells in brain organoids results in stem cell-based models that fail to fully recapitulate the correct microenvironment within the human brain. To address this limitation and to create a vascularized and functionally relevant model, methodologies have been developed to transplant human brain organoids into the rodent brain (Fig. 2B). Notably, the transplantation of brain organoids into rodent brains revealed that, within a physiological context, organoid grafts exhibit progressive neuronal differentiation and maturation, gliogenesis, integration of host microglia, and axon growth in multiple regions of the host brain (Mansour et al, 2018; Revah et al,

2022). The grafting of human neural organoids within an in vivo environment in the animal brain presents a valuable approach for disease modeling under more physiological conditions. In this context, transplanted human neurons mature, activate, and regulate host circuits governing behavior and motor functions (Revah et al, 2022). The integration of brain organoids in an in vivo physiological environment can further facilitate disease modeling by recapitulating cell–cell interactions and axonal projections of patient-derived cells. A very recent study by Zheng and colleagues demonstrated the functional integration of iPSC-derived midbrain organoids into striatal circuits, leading to the restoration of motor function in a mouse model of PD (Zheng et al, 2023). The authors generated organoids from iPSCs and transplanted them into the striatum of 6-hydroxydopamine (6-OHDA)-lesioned immunodeficient mice to investigate grafted organoid efficacy by analyzing functional properties in the transplanted organoids and motor recovery (Zheng et al, 2023). While there is evidence supporting the functionality of DA tissues generated using brain organoid technology, the transplantation of these 3D structures into lesioned mice holds greater significance for advancing fundamental biology rather than attempting to make translational strides in the field of cell therapy. Organoid grafts, particularly those derived from DA neurons, provide a valuable model for investigating advanced stages of neuronal maturation, innervation, and aging (Revah et al, 2022). While in vitro VM organoids provide controlled environments, they do not fully mature or recapitulate the interactions with brain circuitry. Transplanting mature, functional DA neurons into animal models allow researchers to examine their integration into existing neural circuits, long-term survival, and potential to restore function in vivo. This approach may be particularly relevant for modeling neurodegeneration, as it enables the study of how transplanted DA neurons interact with host tissue, respond to external stimuli and stressors, and contribute to functional recovery. Furthermore, these transplantation studies may serve as preclinical platforms for evaluating various therapeutic approaches -whether gene-based, small molecule-driven, or mediated by cellular interventions-designed to improve DA neuron survival, reduce neuroinflammation, and enhance synaptic connectivity (Mansour et al, 2018). By bridging the gap between in vitro models and in vivo systems, organoid transplantation may offer a powerful tool for testing potential PD treatments and advancing our understanding of neurodegeneration. Despite these advances, it is important to highlight that, for stem cell-based therapeutic approaches, transplantation of early DA progenitors derived from 2D cultures remains the most prevalent and promising methodology for generating cells for cell replacement therapy and clinical trials. This approach allows for precise control over the molecular identity of DA progenitors and the generation of homogeneous populations, ensuring consistent and reproducible cell batches for patient transplantation. Once transplanted, these progenitors undergo terminal differentiation and functional maturation within the host brain, which is crucial for their integration and long-term efficacy (Kirkeby et al, 2023; Piao et al, 2021).

## Multi-region brain organoid systems

Recreating and studying cell–cell interactions, neuronal migration, and neuron axonal projection patterns in different brain regions within human brain organoids remains a challenge. An exciting cutting-edge 3D culture approach is based on the generation and fusion of organoids with different regional identities, known as *assembloids* (Fig. 2B) (Bagley et al, 2017). This culture system has already been successfully used to model cortical interneuron migration by assembling ventral and dorsal forebrain organoids (Bagley et al, 2017); (Birey et al, 2017); (Sloan et al, 2018). This system partially overcomes the intrinsic limitation of brain organoids lacking anatomical references, providing models where distinct areas are populated by different, and specific, neuronal populations in co-development (Box 1). Furthermore, this approach allows the study of single isolated interactions between brain regions, such as the nigrostriatal pathway or cortico-thalamic connections, in a simplified, yet fully humanized, system, disassembling the complexity of the mammalian brain piece by piece. Assembloids, formed from patient cells or genetically modified cell lines, can also serve as valuable models for studying disease features (Fig. 2B). For instance, forebrain assembloids derived from individuals with Timothy syndrome, a genetic condition linked to autism and epilepsy, revealed defects in cortical interneuron migration and function. Timothy syndrome neurons exhibit altered saltation frequency and length, associated with changes in actomyosin signaling and GABAergic receptor sensitivity (Birey et al, 2022). Cortico-striatal assembloids generated from patients with Phelan–McDermid syndrome, where the *SHANK3* gene is often lost, demonstrate increased calcium activity and reduced network synchronization (Miura et al, 2020). Significantly, this defect was absent in non-assembled striatal organoids, emphasizing the crucial role of interaction with cortical glutamatergic neurons in uncovering this functional phenotype, which would be challenging to dissect in an animal model (Box 1). These examples underscore the versatility of assembloids in modeling disease-related phenomena, providing insights into intricate cellular interactions and disease mechanisms. The recapitulation of disease features in these models, especially when derived from patient cells is improving our understanding of both neurodevelopmental and neurodegenerative conditions, paving the way for potential therapeutic strategies and drug testing in a fully human context. Assembloids that fuse VM and striatal organoids may provide an effective platform to further drive the differentiation of DA neurons that develop axonal projections toward specific target structures, thus recreating nigrostriatal pathways in a dish. Reumann and colleagues devised organoid models representing the VM, striatum, and cortex - the interconnected regions in the DA system (Reumann et al, 2024). They further developed a method for fusing these distinct organoids, resulting in spatially arranged VM–striatum–cortical organoids, termed MISCOs (ventral MIdbrain–Striatum–COrtical assembloids). Within MISCOs, DA neurons from the midbrain organoid project axons to both the striatum and cortex sides, establishing a high level of DA innervation. Over time, synapses are formed between DA neurons and neurons in the striatum and cortex. In this study, MISCOs were utilized as a platform for a dual purpose: firstly, to assess perturbations in DA circuitry upon cocaine treatment, and secondly, to evaluate the maturation and innervation of DA progenitor cells after injection into a 3D human environment (Reumann et al, 2024). These findings underscore the potential of the assembloid system for studying and interfering with the mechanisms of DA neuron axonal projection patterns. Furthermore, MISCOs represent a versatile and scalable platform to test conditions for cell therapies, including the behavior of distinct cell lines precursors in a 3D human environment that resemble

the targeted transplantation site for current clinical trials. Notably, injected VM-patterned DA progenitors demonstrated maturation and successful innervation of the tissue within MISCOs, providing valuable insights for future therapeutic applications. In a recent study, we have also shown that DA neurons in midbrain organoids fused with striatal organoids (normal targets of the A9 subtype of DA neurons) results in increased and refined A9 identity of the DA neurons in the VM organoid (Fiorenzano et al, 2024).

In addition to assembloids, new organoid-based systems are also emerging for studying cell–cell interactions between different, and anatomically distant, brain areas (Fig. 2). Kirihara and colleagues developed a model of the cerebral tract between two cortical regions by connecting two spheroids through a microchannel where axons could extend and form mutual connections (Kirihara et al, 2019). After characterizing the dynamics of axon growth and fascicle formation between physically confined spheroids, authors exploited this framework to model developmental cerebral tract disorder by knockdown of L1CAM (Kirihara et al, 2019). Furthermore, a similar approach was used to recapitulate the agenesis of corpus callosum caused by ARID1B mutation utilizing patient-derived cerebral organoids, proving once more the value of multi-region organoids in modeling development and diseases (Martins-Costa et al, 2024). Long-range connecting organoid models, or *connectoids*, would be particularly useful for in vitro investigations of DA neuron circuitries, such as the nigrostriatal, mesolimbic and mesocortical pathways, which extend for few millimeters in rodents and few centimeters in humans (Fig. 2B) (Bjorklund and Dunnett, 2007; Garritsen et al, 2023). Beyond models that combine multiple brain regions, emerging microfluidic-based neurodevelopmental platforms offer new possibilities (Box 1). These systems create three-dimensional, lumenal tissue architectures that closely mimic in vivo spatiotemporal cell differentiation and organization, making them powerful tools for studying human neurodevelopment and disease (Xue et al, 2024). By integrating hPSCs with microfluidic neural tube-like structures, these models recapitulate key aspects of neural patterning in both brain and spinal cord regions, as well as along the rostral–caudal and dorsal–ventral axes (Xue et al, 2024). This platform has been used to explore neuronal lineage development, the pre-patterning of axial identities in neural crest progenitors, and the functional roles of neuromesodermal progenitors.

## Conclusions

The development of brain organoid models for studying the DA neurons and their circuitry is opening exciting new avenues beyond traditional animal models or 2D cell cultures. The cutting-edge technologies described in this review enable a more authentic dive into human brain development and provide the potential to precisely model diseases in patient-derived cells or in cells with disease-specific mutations. Despite these breakthroughs, significant challenges remain, and the rapidly evolving field of human stem cell-based models has yet to develop an "ideal" ventral midbrain organoid. Each one of the available approaches today has distinct strengths—and has provided novel insights and exciting discoveries —yet also comes with its own set of limitations. By integrating and critically assessing these diverse models, organoids hold the potential to push the frontiers of our understanding and refine cellular systems. These advancements could extend beyond the midbrain, shedding light on other key regions involved in DAergic

circuitry and advancing our overall conceptual grasp of brain function and disease.

## Peer review information

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

## Acknowledgements

The authors acknowledge financial support from The Crafoord Foundation (20240532) (AF), and Swedish Research Council (2022-01432) (AF).

## Author contributions

**Alessandro Fiorenzano**: Conceptualization; Supervision; Writing—original draft; Writing—review and editing. **Edoardo Sozzi**: Conceptualization; Writing—original draft. **Rahel Kastli**: Writing—original draft. **Maria Roberta Iazzetta**: Writing—original draft. **Andreas Bruzelius**: Writing—original draft. **Paola Arlotta**: Conceptualization; Writing—original draft. **Malin Parmar**: Conceptualization; Writing—original draft.

## Disclosure and competing interests statement

MP is the owner of Parmar Cells AB and co-inventor of the following patents WO2016162747A2, WO2018206798A1, and WO2019016113A1. The remaining authors declare no competing interests.

