## [Peer Review File · The EMBO Journal]

Advances, challenges, and opportunities of human midbrain organoids for modelling of the dopaminergic system

Alessandro FIORENZANO, Edoardo Sozzi, Rahel Kastli, Maria Roberta Iazzetta, Andreas Bruzelius, Paola Arlotta, and Malin Parmar

Corresponding author(s): Alessandro FIORENZANO (alessandro.fiorenzano@med.lu.se)

Review Timeline:

Submission Date:	13th Jan 25
Editorial Decision:	19th Feb 25
Revision Received:	14th Mar 25
Editorial Decision:	5th Jun 25
Revision Received:	6th Jun 25
Accepted:	12th Jun 25

Editor: Ioannis Papaioannou

Transaction Report:

Dear Alessandro, dear Malin,

Thank you again for submitting your Review manuscript (EMBOJ-2025-120151) to The EMBO Journal for our consideration, and for your patience during peer review. Your manuscript has now been seen by two experts in the field, and we have received their comments, which are included below.

I am glad to say that, as you will see, the referees recognize that your Review is of high general interest, novel, and timely. They also identify, however, a number of limitations and list several constructive suggestions for the improvement of the manuscript. Among other concerns, they call for better clarity in some sections as well as citation and discussion of some missing references, provide ideas for additional background information and topics that could be covered/expanded (e.g. points #3-#7 of referee 1, and points #1-#2 of referee 2), and recommend re-organization of the manuscript sections for better clarity and flow.

We have discussed the referees' suggestions in our team, and we largely agree with the majority of the referees' points, as we think that the Review can benefit from the suggested additional content and the recommended re-organization. In light of this input, I would like to invite you to revise your manuscript taking the referees' suggestions on board, and submit the revised version to our online system. Please include in your resubmission a point-by-point response to the referees' reports addressing all raised concerns and comments.

While revising your text, we would kindly request you to also:

- add a list of five relevant keywords after the Abstract
- combine the two References lists in one
- add a "Disclosure and competing interests statement" before the References
- remove the headings "Sozzi et al. - Figure #" from the Figures
- remove the Figures from the main manuscript file and -instead- upload them to our manuscript tracking system as individual high-resolution files when resubmitting your Review; please note that the Figure legends must remain in the manuscript, following the References list
- move the Tables to the end of the manuscript, after the Figure legends.

Regarding the Figures, we think that their design is professional and their content and use adequate and helpful for the reader to follow the text. I would like to kindly ask you to make sure during revision that they are scientifically accurate and that all of their graphical elements are clearly defined/explained in detail in the respective legends. If there are certain aspects of your Figures that are based upon assumptions or where the scientific data remain ambiguous (for example, schematically depicting a presumed direct protein-protein interaction, protein shape or subcellular localization etc.), please add a comment so that we can work with you on an accurate depiction. Please ensure that the directionality and nature of interactions is presented accurately.

Please also note that:

- If the Figures or single panels of the Figures have been adapted from a published Figure, this information must be added to the Figure legend (e.g., 'Adapted from...' or 'Based on...'). I will then discuss with you if a reference and permission will be necessary. Please only re-use Figures or parts of a Figure if this is essential for understanding the concept communicated. Often a reference to a previous paper will suffice.
- If the Figures contain re-used images or elements of images, including schematics, micrographs or photos, please make sure that you have the permission/license to publish them (this also applies to your own previous work, if the journal you previously published in retains copyright). Certain "creative commons" open access licenses, such as CC-BY 4.0, allow re-use without additional formal permissions. All re-used material must be explicitly cited.
- If you use an image database for scientific iconography (such as BioRender), please acknowledge it in the respective Figure legends and make sure that you have a license that allows for publication in an academic journal.

Please also note that as part of the EMBO publications' Transparent Editorial Process, The EMBO Journal publishes online a Peer Review File along with each accepted manuscript. This File will be published in conjunction with your Review article and will include the referee report, your point-by-point response, and all pertinent correspondence relating to the manuscript. You can opt out of this by letting the editorial office know (contact@embojournal.org). If you do opt out, the Peer Review File link will point to the following statement: "No Peer Review File is available with this article, as the authors have chosen not to make the review process public in this case."

We look forward to your revised Review article addressing the above points as soon as possible. Please let us know if you have any questions or comments you would like to discuss with me. When you are ready to re-submit your revision, please use the link:

<https://emboj.msubmit.net/cgi-bin/main.plex>.

Best wishes,

Ioannis

Referee #1:

In this manuscript, the authors reviewed the generation of midbrain organoids and disease modeling with these organoids. The brain organoid field is rapidly evolving and a specific review focuses on the midbrain is useful. The authors need to articulate across the whole manuscript on why organoids, instead of previous 2D cultures. The organization and flow of the manuscript can be significantly enhanced. Here are a number of specific suggestions for the authors to improve their piece.

1. The title is about human midbrain organoids, which do not only include DA neurons, but other cell types. This is the main point of using organoid model, instead of more controlled 2D differentiation.
2. In the first part of historically development of VM organoids (page 3 first line), all three papers cited were for 2D differentiation. They author should mention and cite the first real 3D model (Qian et al Cell 2016) and describe the advancement in the field chronologically, instead of mix them all together.
3. The authors then go straight to PD in the next section. They need to discuss other aspect of neural development, in addition to DA neurons.
4. In discussing PD modeling using VM organoids, the authors need to be clear that VM organoid is a developmental model, not for age-dependent neurological disorders. What aspects can be modeled and used for drug screen and what are the additional developments need to be made to expand the usefulness of VM models for PD? Also authors should discuss what developmental questions can be addressed using VM organoids.
5. In my view, it may better to talk about advancement in midbrain organoid technology: lessons from forebrain human models first, and then go to PD and drug screen.
6. The authors discussed the interaction of DA neurons with other brain regions and should cite specific examples of assembloid studies involving VM organoids. In addition to assembloids which put two brain region artificially together, there are new models that can generate the whole human neural tube, including forebrain, midbrain, hindbrain and spinal cord aligned naturally together (Xue et al. Nature 2024). This should be included in the multi-region organoid systems section.
7. In the transplantation section, the authors can discuss how transplantation can be used to overcome maturation of DA neurons in dish for modeling neurodegeneration in animals in vivo and test therapeutic strategies as a preclinical model.

Referee #2:

Fiorenzano et al gives an overview of the human midbrain organoids model. The review is timely in summarizing the recent advances in midbrain organoid field and offers future challenges and possible solutions based on the more advanced forebrain organoids. Overall, writing is excellent, but the manuscript needs major re-organization in its sections.

1. It is unclear whether the authors describe midbrain organoids, or ventral midbrain organoids. The title is "Human midbrain organoids:...", but all the focus is ventral midbrain. Authors need to change the title. Additionally, in the introduction, authors should define the midbrain, describe dorsal and ventral midbrain, and then focus on ventral midbrain.
2. As described in the manuscript, ventral midbrain is involved in a variety of neuropsychiatric disorders in addition to PD. Authors should expand this, if they want to emphasize it. In current format, abstract is mainly on PD but neuropsychiatric disorder-related contents are importantly described. This may be an important content to add in using the midbrain organoids.
3. Sections need re-organized. "ModelingVM Development" section is composed of a couple of irrelevant subsections, such as "Drug screening in VM organoids". Authors should have first subsection for developmental biology of midbrain and in vitro formation of midbrain organoids. There should be a section of application of midbrain organoids, which should include modeling PD, maybe an additional "neuropsychiatric disorders", and then drug screening. The whole section of "Advancement in midbrain organoid technology: Lessons..." seem out of context for the manuscript. This section should be Text Box with short summary. There are already a large number of reviews for forebrain organoid models. The section should be "Challenge and improvement of midbrain organoids", and give subsections as given in the manuscript.
4. It is recommended to include one more table that describes the use of midbrain organoids to model midbrain disorders,

including PD and other relevant midbrain disorders.

5. References are not placed well. There are two bibliography sections, and throughout the manuscript, the format of some references is inconsistent.

Referee#1:

In this manuscript, the authors reviewed the generation of midbrain organoids and disease modeling with these organoids. The brain organoid field is rapidly evolving and a specific review focuses on the midbrain is useful. The authors need to articulate across the whole manuscript on why organoids, instead of previous 2D cultures. The organization and flow of the manuscript can be significantly enhanced. Here are a number of specific suggestions for the authors to improve their piece.

We sincerely appreciate the reviewer's recognition of the relevance of a focused review on midbrain organoids and for the constructive feedback on improving the manuscript. Below, we provide a point-by-point response detailing the revisions made in accordance with the reviewer's suggestions.

1. The title is about human midbrain organoids, which do not only include DA neurons, but other cell types. This is the main point of using organoid model, instead of more controlled 2D differentiation.

We have now clarified in the manuscript (page 2, line 24) and acknowledge that one of the key advantages of midbrain organoids over 2D cultures is their ability to better recapitulate the cellular diversity of the human VM.

2. In the first part of historically development of VM organoids (page 3 first line), all three papers cited were for 2D differentiation. They author should mention and cite the first real 3D model (Qian et al Cell 2016) and describe the advancement in the field chronologically, instead of mix them all together.

We apologize for the confusion. We have now revised this part of the manuscript to describe the field's progression chronologically and have appropriately cited Qian et al. (Cell, 2016) as the first significant 3D model for VM organoids. This revision now clarifies the transition from 2D differentiation systems to the establishment of 3D organoid models and better reflects the evolution of this area of research.

3. The authors then go straight to PD in the next section. They need to discuss other aspect of neural development, in addition to DA neurons.

We thank the reviewer for this valuable suggestion. In the revised version, we have expanded the discussion to include additional aspects of neural development beyond DA neurons. Specifically, we now highlight the role of non-dopaminergic cell populations, such as GABAergic and glutamatergic neurons, as well as the critical contributions of glial cells, including astrocytes, oligodendrocytes, and microglia, in midbrain development and circuit formation (page 3 and 4).

4. In discussing PD modeling using VM organoids, the authors need to be clear that VM organoid is a developmental model, not for age-dependent neurological disorders. What aspects can be modeled and used for drug screen and what are the additional developments need to be made to expand the usefulness of VM models for PD? Also authors should discuss what developmental questions can be addressed using VM organoids.

In the revised version of the review, we have clarified that VM organoids primarily serve as developmental models rather than fully recapitulating age-dependent neurodegenerative disorders like PD. We now explicitly discuss the aspects of PD that can be modeled using VM organoids, such as early-stage neurodevelopmental deficits, genetic predispositions (page 5 and 6) and also the concept of direct conversion where age is maintained. Finally, we explored the broader developmental questions that VM organoids can address, including the investigation of dopaminergic subtypes with distinct functions and innervation targets within the midbrain (page 3 line 33).

5. In my view, it may better to talk about advancement in midbrain organoid technology: lessons from forebrain human models first, and then go to PD and drug screen.

We appreciate the reviewer's suggestion regarding the restructuring of the manuscript. In the revised version, we first discuss the development of midbrain organoids before transitioning into the Biomedical Applications subsection, which includes PD modeling and drug screening, ensuring a more coherent narrative. Additionally, we have converted the section titled "Advancements in Midbrain Organoid Technology: Lessons from Forebrain Human Models" into a text box to maintain the logical flow of the review.

6. The authors discussed the interaction of DA neurons with other brain regions and should cite specific examples of assembloid studies involving VM organoids. In addition to assembloids which put two brain region artificially together, there are new models that can generate the whole human neural tube, including forebrain, midbrain, hindbrain and spinal cord aligned naturally together (Xue et al. Nature 2024). This should be included in the multi-region organoid systems section.

We have included specific examples of assembloid studies involving VM organoids to better highlight the interaction of dopaminergic neurons with other brain regions (Reumann et al., Nature Meth 2023, Fiorenzano, Strom et al.

PNAS 2024). Additionally, we have expanded the section on multi-region organoid systems to incorporate emerging models that can generate the entire human neural tube, including the forebrain, midbrain, hindbrain, and spinal cord, as described by Xue et al. Nature 2024 (page 12-13).

7. In the transplantation section, the authors can discuss how transplantation can be used to overcome maturation of DA neurons in dish for modeling neurodegeneration in animals in vivo and test therapeutic strategies as a preclinical model. We have incorporated a discussion on how transplantation can be used to overcome the limitations of DA neuron maturation in vitro (page 10-11).

Referee#2:

Fiorenzano et al gives an overview of the human midbrain organoids model. The review is timely in summarizing the recent advances in midbrain organoid field and offers future challenges and possible solutions based on the more advanced forebrain organoids. Overall, writing is excellent, but the manuscript needs major re-organization in its sections.

We appreciate the positive remarks regarding the overview of the human midbrain organoid model and the timeliness of the review in summarizing recent advances in the field. We also value the insightful suggestion on the manuscript's organization and now submit a reorganized version based on both reviewers comments.

1. It is unclear whether the authors describe midbrain organoids, or ventral midbrain organoids. The title is "Human midbrain organoids:...", but all the focus is ventral midbrain. Authors need to change the title. Additionally, in the introduction, authors should define the midbrain, describe dorsal and ventral midbrain, and then focus on ventral midbrain.

We have revised the title to more accurately reflect the focus on ventral midbrain organoids. Additionally, we have updated the introduction to clearly distinguish between the dorsal and ventral midbrain, with a more detailed emphasis on the ventral midbrain (page 1).

2. As described in the manuscript, ventral midbrain is involved in a variety of neuropsychiatric disorders in addition to PD. Authors should expand this, if they want to emphasize it. In current format, abstract is mainly on PD but neuropsychiatric disorder-related contents are importantly described. This may be an important content to add in using the midbrain organoids.

We thank the reviewer for this suggestion. We have expanded the discussion on the use of VM organoids to model neurodegenerative diseases, including Progressive Supranuclear Palsy (PSP) and Huntington's Disease (HD). However, VM application in modeling neuropsychiatric disorders remains limited, primarily due to the challenges in generating specific sub-regions of the VM in vitro, which are essential for accurately recapitulating these dysfunctions. This issue is now further addressed on page 5, line 30.

3. Sections need re-organized. "ModelingVM Development" section is composed of a couple of irrelevant subsections, such as "Drug screening in VM organoids". Authors should have first subsection for developmental biology of midbrain and in vitro formation of midbrain organoids. There should be a section of application of midbrain organoids, which should include modeling PD, maybe an additional "neuropsychiatric disorders", and then drug screening. The whole section of "Advancement in midbrain organoid technology: Lessons..." seem out of context for the manuscript. This section should be Text Box with short summary. There are already a large number of reviews for forebrain organoid models. The section should be "Challenge and improvement of midbrain organoids", and give subsections as given in the manuscript.

Thank you for your insightful suggestion regarding the reorganization of the manuscript sections. In response, we have restructured the content as follows:

1. We have created a new first subsection titled **"In Vitro Modeling of Human Ventral Midbrain Development Using Midbrain Organoids,"** which focuses on the VM development and the in vitro formation of midbrain organoids.
2. The following subsection, **"VM Organoids for Biomedical Applications,"** addresses the various applications of midbrain organoids, including modeling Parkinson's Disease, neuropsychiatric disorders, and their use in drug screening.
3. Lastly, we have included a new section titled **"Challenges and Progress in Midbrain Organoid Research,"** which highlights the key challenges and developments in midbrain organoid technology. We

have moved “**Advancement in Midbrain Organoid Technology: Lessons Learned**” into **Text Box 1**, providing a concise summary of advancements in the field.

4. It is recommended to include one more table that describes the use of midbrain organoids to model midbrain disorders, including PD and other relevant midbrain disorders.

Thank you for the suggestion to include an additional table on midbrain organoids modeling disorders like PD. While we initially intended to include this, we found that the available research on using midbrain organoids for neuropsychiatric disorders is still limited. As a result, we decided not to include a table at this time and have instead addressed the emerging findings and challenges in the text (page 5).

5. References are not placed well. There are two bibliography sections, and throughout the manuscript, the format of some references is inconsistent.

Thank you for pointing out the issues with the references. We have reorganized the bibliography to ensure there is only one section.

Dear Alessandro,

Thank you again for submitting your revised Review manuscript (EMBOJ-2025-120151R) to The EMBO Journal for our consideration, and for your patience. Once again, please accept my apologies for the slow process on this occasion, but we have now received expert input on the revised text -which, I am glad to say, was very supportive- and we have also read and discussed the revised text in detail in our team, and we have a number of suggestions for minor edits we would like to make. Please find attached an edited copy of your review article with comments and tracked changes from our team -which we invite you to take on board while preparing the final version of your review manuscript- and submit the new version to our online system as soon as possible.

While revising your text, we would kindly also request you to:

- indicate the corresponding author (also providing an institutional e-mail address) on the title page of the manuscript
- make sure that the funding information provided in our manuscript tracking system (eJP) is identical to that included in the Acknowledgements section of the manuscript (information on "Swedish Research Council (2022-01432) and the Royal Physiographic Society in Lund (44435 and 43374)" is currently missing in eJP)
- add callout(s) for Fig. 2B where appropriate in the main text
- make sure that all Text Boxes are also called out in the main text.

Please include the Text Boxes at the end of the main text, before the list of References, in the final version of your manuscript.

I would also like to remind you that if the Figures contain re-used images or elements of images, including schematics, micrographs or photos, please make sure that you have the permission/license to publish them (this also applies to your own previous work, if the journal you previously published in retains copyright). Certain "creative commons" open access licenses, such as CC-BY 4.0, allow re-use without additional formal permissions. All re-used material must be explicitly cited. If you use an image database for scientific iconography (such as BioRender), please acknowledge it in the respective Figure legends and make sure that you have a license that allows for publication in an academic journal.

Please also note that as part of the EMBO publications' Transparent Editorial Process, The EMBO Journal publishes online a Peer Review File along with each accepted manuscript. This File will be published in conjunction with your review article and will include the referee reports, your point-by-point response, and all pertinent correspondence relating to the manuscript. You can opt out of this by letting the editorial office know (contact@embojournal.org). If you do opt out, the Peer Review File link will point to the following statement: "No Peer Review File is available with this article, as the authors have chosen not to make the review process public in this case."

We look forward to the final version of your review article addressing the above points as soon as possible. Please let us know if you have any questions or comments you would like to discuss with me. When you are ready to submit your revision, please use the link:

<https://emboj.msubmit.net/cgi-bin/main.plex>.

With thanks and best wishes,

Ioannis

All editorial and formatting issues were resolved by the authors.

Dear Alessandro, dear Malin,

Congratulations on an excellent work! I am very pleased to inform you that your review article has been accepted for publication in The EMBO Journal. Thank you very much for comprehensively addressing the initially raised referees' concerns and all editorial requests for changes.

Your manuscript will be processed for publication by EMBO Press. It will be copy edited and you will receive page proofs prior to publication.

Since this is a commissioned review article, we are happy to waive (100%) the publication charge (APC). When you are contacted by Springer Nature Author Services to complete licensing and payment information, please use the token below for getting the fee waived:

[the token has been removed]

If you have any questions, please do not hesitate to contact the Editorial Office. Thank you very much for your contribution to The EMBO Journal. Working with you has been a pleasure!

All the best,

Ioannis
